# Overcoming the Pitfalls of Vision-Language Model Finetuning for OOD Generalization

**Yuhang Zang**[*1], **Hanlin Goh**[2], **Josh Susskind**[2], **Chen Huang**[2]

[1]Nanyang Technological University  [2]Apple Inc.
zang0012@ntu.edu.sg  {hanlin,jsusskind,chen-huang}@apple.com

## Abstract

Existing vision-language models exhibit strong generalization on a variety of visual domains and tasks. However, such models mainly perform zero-shot recognition in a closed-set manner, and thus struggle to handle open-domain visual concepts by design. There are recent finetuning methods, such as prompt learning, that not only study the discrimination between in-distribution (ID) and out-of-distribution (OOD) samples, but also show some improvements in both ID and OOD accuracies. In this paper, we first demonstrate that vision-language models, after long enough finetuning but without proper regularization, tend to overfit the known classes in the given dataset, with degraded performance on unknown classes. Then we propose a novel approach OGEN to address this pitfall, with the main focus on improving the OOD GENeralization of finetuned models. Specifically, a class-conditional feature generator is introduced to synthesize OOD features using just the class name of any unknown class. Such synthesized features will provide useful knowledge about unknowns and help regularize the decision boundary between ID and OOD data when optimized jointly. Equally important is our adaptive self-distillation mechanism to regularize our feature generation model during joint optimization, *i.e.*, adaptively transferring knowledge between model states to further prevent overfitting. Experiments validate that our method yields convincing gains in OOD generalization performance in different settings. Code: https://github.com/apple/ml-ogen.

## 1 Introduction

Large-scale pre-trained vision-language models like CLIP (Radford et al., 2021) demonstrate promising generalizability on various visual domains and tasks in the real world. However, their zero-shot in-distribution (ID) performance can be limited for some downstream datasets. Also due to their zero-shot evaluation in a closed-set manner (*i.e.*, to match input image to a predefined set of classes), vision-language models often struggle to handle the out-of-distribution (OOD) samples from novel classes. Such shortcomings create major safety risks in the open domain that often require capabilities of OOD detection and/or accurate identification of both novel and seen classes.

Some recent works attempt to improve the *zero-shot* OOD detection performance of existing vision-language models, either by simple softmax scaling (Ming et al., 2022) or training an extra text generator (Esmaeilpour et al., 2022). Alternatively, Fort et al. (2021) first show the promise of CLIP models *finetuned* on an ID dataset. Encouragingly both ID and OOD accuracies are improved after finetuning. Parameter-efficient finetuning methods, such as prompt learning (Zhou et al., 2022a;b) or adaptor tuning (Zhang et al., 2022), illustrate similar benefits without heavy training.

Despite the success of prior finetuning methods, we found from our extensive benchmarking that finetuning on ID datasets is prone to overfitting (Fig. 1(b)). More specifically, we observed that models after long enough finetuning but without proper regularization, tend to overfit the known classes in the given dataset, with inferior generalization on unknown classes. Unfortunately, an explicit regularization mechanism has not been explored in literature to address this pitfall, and simple

---

[*]Work done while interning at Apple.

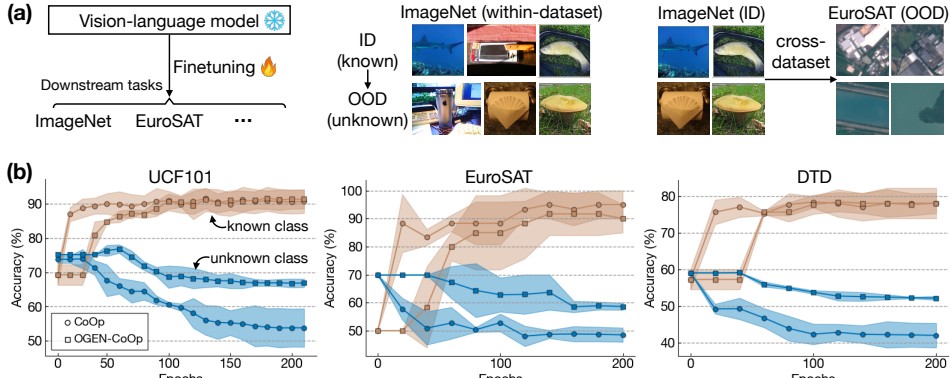

Figure 1: **(a)** We study OOD generalization when finetuning the vision-language model CLIP on various downstream tasks. We consider both within-dataset generalization where one dataset has **ID** vs. **OOD** (or **known** vs. **unknown**) class splits for finetuning and evaluation respectively, and the more challenging cross-dataset generalization setting. More clarifications on the problem definition in Appendix A. **(b)** Examples of within-dataset generalization: we show learning curves of the prompt learning method CoOp (Zhou et al., 2022a) that finetunes CLIP for long enough (200 epochs) on three datasets (more in Appendix B). Apparently, CoOp overfits the known classes of each dataset with notable accuracy drop on the unknowns. Our proposed method OGEN largely reduces such overfitting through effective regularization.

regularization strategies like early stopping seem insufficient. E.g. in Fig. 1(b), it is difficult to find an early model checkpoint with good trade-off between the known and unknown class performance.

One main challenge of effective model regularization is the missing knowledge about unknowns. Such knowledge could actually offer useful supervision signals to avoid overconfident predictions on OOD data. In this paper, we propose a novel method that features 1) image feature synthesis for unknown classes and 2) an unknown-aware finetuning algorithm with effective model regularization. The goal is to improve OOD generalization without hurting the ID performance of finetuned models. To synthesize unknown features, we introduce a class-conditional feature generator: *i.e.*, generating image features just given the name of an unknown class. This is made possible by CLIP's well-aligned image-text feature spaces. Our feature generator is implemented by a lightweight attention module, with an "extrapolating bias" on the unknown classes. It generalizes well to "unknown unknowns" and hence can model the complex distributions of visual classes in the open domain. Then we use both the ID and synthesized OOD data for joint optimization, leading to a better regularized decision boundary. Another contribution is an adaptive self-distillation mechanism that regularizes our feature generator to further reduce overfitting during joint optimization. The idea is to find an adaptive teacher model of the feature generator from historical training epochs (with less overfitting) to guide optimization at the current epoch (student model, often with more overfitting).

Our overall approach OGEN is applicable to different finetuning methods *e.g.*, (Zhou et al., 2022a;b; Jia et al., 2022) for CLIP-like models. OGEN is shown to consistently improve their OOD generalization performance (by up to absolute 18.77%) under two settings: within-dataset (base-to-new class) generalization and cross-dataset generalization. Summarizing, our **main contributions** are:

- Provide the first *comprehensive* study on OOD generalization that unveils the pitfalls of finetuning methods (based on prompt learning) for vision-language models.
- A class-conditional feature generator to synthesize OOD data for effective regularization.
- Adaptive self-distillation on our feature generator to further reduce overfitting.

## 2 RELATED WORK

**Vision-Language Models.** Recent large-scale vision-language models like ViLT (Kim et al., 2021) and PaLI (Chen et al., 2023) simply consume image-and-text features for multimodal downstream tasks with remarkable performance. Another popular paradigm used in CLIP (Radford et al., 2021) and ALIGN (Jia et al., 2021) contrastively aligns image and text encoders. These contrastive models are trained on massive web-scale image-text pairs, also showing strong adaptability to a range of

downstream tasks, such as semantic segmentation (Zang et al., 2022a; Ghiasi et al., 2021) and video classification (Qian et al., 2022). Numerous follow-up works (Li et al., 2022; Zhou et al., 2022a) aim to improve CLIP-like models in data efficiency or generalization. However, the zero-shot performance on some tasks can still be limited for existing vision-language models. Hu et al. (2023) found that they make different kinds of errors, *e.g.*, PaLI is erroneous at tail visual concepts while CLIP may fail for common ones. This paper mainly studies and improves the generalization of finetuned CLIP models, but our approach is model-agnostic and thus applicable to other vision-language models as well.

**Finetuning methods** have been studied to improve the downstream performance of vision-language models over their zero-shot counterparts. Fort et al. (2021) showed that after finetuning the CLIP model on datasets of interest, both the ID and OOD generalization performance will be improved. More parameter-efficient finetuning methods are popularized in recent years. In particular, prompt learning focuses on learning visual (Jia et al., 2022), textual (Zhou et al., 2022a;b; Yao et al., 2023; Wang et al., 2023; Shu et al., 2023; Khattak et al., 2023b) or multi-modal Zang et al. (2022b); Khattak et al. (2023a) prompts, while adaptor tuning (Zhang et al., 2022) optimizes feature representations with the model backbone kept frozen. In this paper, we first unveil the overfitting issue of recent finetuning methods, and then propose a new regularization method to prevent overfitting. Our approach is orthogonal to the finetuning research, and shows consistent gains over various finetuning baselines.

**Outlier synthesis** proves effective for model regularization in the absence of OOD data. Previous methods rely on GANs (Lee et al., 2018) to synthesize outlier images. More recent methods like VOS (Du et al., 2022) directly synthesize virtual features which allows greater flexibility. Tao et al. (2023) propose non-parametric outlier synthesis, without the restrictive Gaussian assumption on feature distributions in VOS. Here we present a new feature synthesis method that has the same format as the CLIP framework and hence facilitates multimodal regularization. Specifically, given the name of an unknown class, we synthesize its example features in a generalizable way.

**Model distillation** techniques transfer knowledge from a teacher model to student models, *e.g.*, from a large model to its efficient counterparts (Hinton et al., 2015) or from a weakly augmented model to the strongly augmented (Sohn et al., 2020). Here we aim to reduce overfitting for unseen classes and propose to distill knowledge from early to current epochs (*i.e.*, self-distillation). Specifically, we extend Mean teacher (Tarvainen & Valpola, 2017) to an adaptive localized one with suitable teacher curriculum. In the vision-language domain, our approach differs from distillation into smaller models (Li et al., 2023) or towards various downstream tasks (Gu et al., 2022; Dai et al., 2022; Mal et al., 2022). Our approach is also orthogonal (and applicable) to recent distillation frameworks for improved multimodal *pretraining* (Dong et al., 2023; Li et al., 2021; Zhong et al., 2022).

## 3 METHODOLOGY

### 3.1 PRELIMINARIES

**CLIP** (Radford et al., 2021) is the vision-language model that we mainly study in this paper, although our study is applicable to other popular models. CLIP consists of an image encoder $\phi$ and a text encoder $\psi$, which map the image and text inputs into a joint feature space. The CLIP training aims at aligning the image and text modalities by maximizing their feature similarity. Given an input image $x$ that belongs to one of the classes $Y = \{y_1, y_2, ..., y_C\}$, the image encoder $\phi$ first extracts image features $z = f_\phi(x) \in \mathbb{R}^d$. To obtain the corresponding text features $w_{c \in \{1,...,C\}}$, all the given class names can be fed into a fixed prompt template {a photo of a [CLASS]}, leading to text descriptions $A$ which are further encoded by $\psi$ into the text embeddings $W = f_\psi(A) \in \mathbb{R}^{d \times C}$ (hence $w_c = W_{:,c}$). The image-text alignment is optimized based on the cosine feature similarity:

$$p(y = c \mid x) = \frac{\exp\left(\cos\left(w_c, z\right)/\tau\right)}{\sum_{i=1}^{C} \exp\left(\cos\left(w_i, z\right)/\tau\right)}, \tag{1}$$

where $\tau$ is the temperature. A larger cosine score often indicates stronger image-text alignment in their underlying semantics.

**Prompt Learning.** For efficient model finetuning on downstream tasks, recent prompt learning approaches like CoOp (Zhou et al., 2022a) replace the aforementioned fixed prompts with learnable

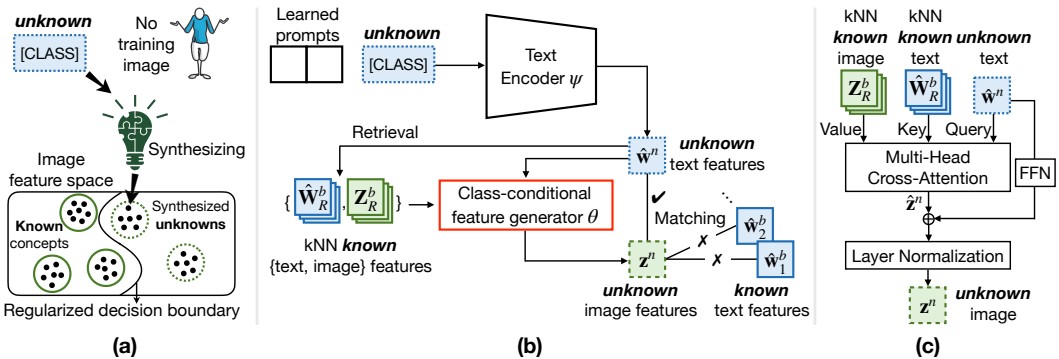

Figure 2: **(a)** To improve OOD generalization, we propose to gain knowledge of unknown classes by directly synthesizing their image features. This helps to learn a more reliable decision boundary between known and unknown classes in the feature space. **(b)** Prompt learning based on discriminating both the known and synthesized unknown features (from our class-conditional feature generator $\theta$, see details in text). **(c)** Implementation of $\theta$ using a lightweight attention module.

ones $\boldsymbol{V} = [\boldsymbol{v}_1, \boldsymbol{v}_2, \ldots, \boldsymbol{v}_L] \in \mathbb{R}^{d \times L}$ where $L$ is the prompt length. Then the text encoder $\psi$ of CLIP will be able to convert the learned prompts $\boldsymbol{V}$ (together with $\boldsymbol{Y}$) into adapted text embeddings $\hat{\boldsymbol{W}} = f_\psi([\boldsymbol{V}, \boldsymbol{Y}]) \in \mathbb{R}^{d \times C}$. Note $\boldsymbol{V}$ is learned on each downstream task using the task-specific loss. The image encoder $\phi$ and text encoder $\psi$ of CLIP are kept frozen during prompt learning.

## 3.2 CLASS-CONDITIONAL FEATURE GENERATOR

As shown in Fig. 1(b), the "prompt-tuned" CLIP model tends to overfit the known classes (*aka* base classes $\boldsymbol{Y}^b = \{\boldsymbol{y}_1, \boldsymbol{y}_2, ..., \boldsymbol{y}_{C_b}\}$) from the downstream task, while OOD generalization on unknown classes (*aka* new classes $\boldsymbol{Y}^n$ with $|\boldsymbol{Y}^n| = C_n$) will deteriorate. To reduce overfitting, one might choose model regularization strategies, which will inevitably suffer from the missing knowledge about unknowns. Moreover, the potential number of unknown classes $C_n$ is huge and $C_n \gg C_b$. Hence it is very challenging to model their complex distributions for effective regularization.

Here we make one step towards gaining knowledge of unknowns in a class-conditional manner, in order to provide supervision signals for the vast space of unknown data. Given a textual description or simply the class name of *any* unknown class, we aim to synthesize the class example *features* without seeing labeled instances (Fig. 2(a)), leveraging the well-aligned image-text feature spaces of CLIP. Such synthesized image features will then facilitate learning a regularized decision boundary between known and unknown classes, leading to improved OOD generalization capabilities.

In early experiments, we found that directly generating OOD image features out of class names is hard due to the highly non-linear and high-dimensional nature of the former. This is similarly observed in those strong cases of OOD generalization in (Abbe et al., 2023), where the manifold embeddings are typically nonlinear and, more critically, part of the distribution domain is entirely unseen at training. It is proved that successful learning under such extreme distribution shift leads to extrapolating solutions since memorization is voided on the unseen domain. Following the "extrapolating bias" on the unknown, we reframe our feature synthesis problem as an easier one — extrapolating from the most similar classes of the seen data, *e.g.*, to generate features of the unknown class *raccoon* by extrapolating features of the similar training classes like *cat* and *bear*.

More specifically, for prompt learning, given the learned prompts and one unknown [CLASS] from the open set $\boldsymbol{Y}^n$, we first obtain the corresponding text features $\hat{\boldsymbol{w}}^n \in \mathbb{R}^d$ through the text encoder $\psi$ of CLIP. Then we find for $\hat{\boldsymbol{w}}^n$ its kNN classes from the entire set of text features of known classes $\hat{\boldsymbol{W}}^b \in \mathbb{R}^{d \times C_b}$, resulting in $\hat{\boldsymbol{W}}_R^b \in \mathbb{R}^{d \times K}$ where $R$ is the neighbor set with $|R| = K$. From each of the kNN classes, we randomly sample only one class example and obtain its text-aligned image features from the image encoder $\phi$, leading to the same number of $K$ image feature vectors $\boldsymbol{Z}_R^b \in \mathbb{R}^{d \times K}$. Our goal is to train a class-conditional feature generator $f_\theta(\hat{\boldsymbol{w}}^n, \hat{\boldsymbol{W}}_R^b, \boldsymbol{Z}_R^b)$ that can synthesize unknown image features conditioned on the text features $\hat{\boldsymbol{w}}^n$ of an unknown class and auxiliary text/image features $(\hat{\boldsymbol{W}}_R^b, \boldsymbol{Z}_R^b)$ of kNN known classes, see Fig. 2 (b).

**Remarks.** To retrieve semantically similar kNN classes $\hat{\boldsymbol{W}}_R^b$ from $\hat{\boldsymbol{W}}^b$, we choose to use the cosine similarity score between the text features (not image features) of class pairs. Then the kNN retrieval

process can be formally defined as:

$$\underset{R \subset \{1,\ldots,C_b\}:|R|=K}{\arg\max} \sum_{i \in R} \cos\left(\hat{\boldsymbol{w}}^n, \hat{\boldsymbol{w}}_i^b\right), \ where \ \hat{\boldsymbol{w}}_i^b = \hat{\boldsymbol{W}}_{:,i}^b. \tag{2}$$

On another note, our empirical study shows that the one random example sampled from each kNN class is enough for assisting new feature generation. Such randomness encourages the diversity of the synthesized features for new classes.

**Extrapolating per class.** Recall the tuple $(\hat{\boldsymbol{W}}_R^b, \boldsymbol{Z}_R^b)$ consists of $K$ text and image feature vectors respectively (one for each similar known class). One straightforward feature synthesis method for an unknown class (with text features $\hat{\boldsymbol{w}}^n$) is to extrapolate each image feature vector in $\boldsymbol{Z}_R^b$ based on some notion of similarity with $\hat{\boldsymbol{w}}^n$, leading to a total of $K$ extrapolated image features from $K$ known classes (*e.g.*, *cat→raccoon*, *bear→raccoon*,...). The similarity notion can be well learned by Multi-Head Cross-Attention (MHCA) that operates on the triplets of queries, keys and values $(\hat{\boldsymbol{w}}^n, \hat{\boldsymbol{W}}_R^b, \boldsymbol{Z}_R^b)$. This way, we can effectively take into account the similarity between the unknown class and each known class in $R$ as well as all other between-class similarities. Summarizing, the matrix form of our "extrapolating-per-class" scheme is given as:

$$\boldsymbol{Z}^n = \mathtt{LN}(\boldsymbol{Z}_R^b + \hat{\boldsymbol{Z}}^n) \in \mathbb{R}^{d \times K}, \ \ \hat{\boldsymbol{Z}}^n = \mathtt{MHCA}(\hat{\boldsymbol{w}}^n \cdot \mathbf{1}_K^\top, \hat{\boldsymbol{W}}_R^b, \boldsymbol{Z}_R^b) \in \mathbb{R}^{d \times K}, \tag{3}$$

where $\hat{\boldsymbol{Z}}^n$ are the learned feature residuals when extrapolating each of the $K$ known classes. $\mathtt{LN}$ denotes layer normalization. Obviously, our feature generator $\theta$ is lightweight with only one $\mathtt{MHCA}$ layer and one $\mathtt{LN}$ layer. The simplicity benefits from the "extrapolating bias" in our generator design.

Finally, we use the synthesized features $\boldsymbol{Z}^n$ to regularize prompt learning and perform joint discrimination of $C_b$ known and $C_n$ unknown class features. The objective of maximizing the image-text alignment in Eq. (1) now becomes:

$$p(y = c \mid \boldsymbol{Z}^n) = \frac{1}{K} \sum_{k=1}^{K} \frac{\exp\left(\cos\left(\hat{\boldsymbol{w}}_c, \boldsymbol{z}_k^n\right)/\tau\right)}{\sum_{i=1}^{C_b+C_n} \exp\left(\cos\left(\hat{\boldsymbol{w}}_i, \boldsymbol{z}_k^n\right)/\tau\right)}, \forall c \in \{1, \ldots, C_b + C_n\}, \tag{4}$$

where $\hat{\boldsymbol{w}}_c = [\hat{\boldsymbol{W}}^b, \hat{\boldsymbol{W}}^n]_{:,c}$ and $\boldsymbol{z}_k^n = \boldsymbol{Z}_{:,k}^n$. Note under the "extrapolating-per-class" scheme, we have synthesized $K$ image features for the same unknown class. We simply aggregate them at the score level when computing the cosine feature similarity score in Eq. (4).

**Extrapolating jointly** is a more collaborative approach for new feature synthesis. As the name hints, we extrapolate a *single* image feature vector $\boldsymbol{z}^n$ from all the kNN known class features $(\hat{\boldsymbol{W}}_R^b, \boldsymbol{Z}_R^b)$, based on the cross attention against $\hat{\boldsymbol{w}}^n$:

$$\boldsymbol{z}^n = \mathtt{LN}(\mathtt{FFN}(\hat{\boldsymbol{w}}^n) + \hat{\boldsymbol{z}}^n) \in \mathbb{R}^d, \ \ \hat{\boldsymbol{z}}^n = \mathtt{MHCA}(\hat{\boldsymbol{w}}^n, \hat{\boldsymbol{W}}_R^b, \boldsymbol{Z}_R^b) \in \mathbb{R}^d, \tag{5}$$

where $\hat{\boldsymbol{z}}^n$ is the residual image feature vector, while text features $\hat{\boldsymbol{w}}^n$ are projected into the image feature space via a two-layer fully connected feed-forward network $\mathtt{FFN}$. Note $\mathtt{FFN}(\hat{\boldsymbol{w}}^n)$ could be replaced by some anchor point directly searched in the image feature space, *e.g.*, a weighted average of kNN image features from $\boldsymbol{Z}_R^b$. However, searching is a hard problem itself and learning an explicit text-to-image feature mapping works consistently better in our experiments. Fig. 2 (c) summarizes the overall network architecture, and the objective function in Eq. (4) could be updated as:

$$p(y = c \mid \boldsymbol{z}^n) = \frac{\exp\left(\cos\left(\hat{\boldsymbol{w}}_c, \boldsymbol{z}^n\right)/\tau\right)}{\sum_{i=1}^{C_b+C_n} \exp\left(\cos\left(\hat{\boldsymbol{w}}_i, \boldsymbol{z}^n\right)/\tau\right)}, \forall c \in \{1, \ldots, C_b + C_n\}. \tag{6}$$

**Remarks.** Our ablation study (Table 4) shows that "extrapolating jointly" (**our default approach**) is better than "extrapolating per class" at synthesizing useful unknown features for joint optimization. We train our class-conditional feature generator using the "known" and "unknown" class splits from the training set of downstream tasks. Fig. 3 demonstrates the ability of our feature generator to generalize to "unknown unknowns" during testing, with faithful image feature synthesis.

## 3.3 Adaptive Self-Distillation

Optimizing both known and synthesized unknown features generally improves OOD generalization and oftentimes the ID performance too. However, that does not take into account the optimization

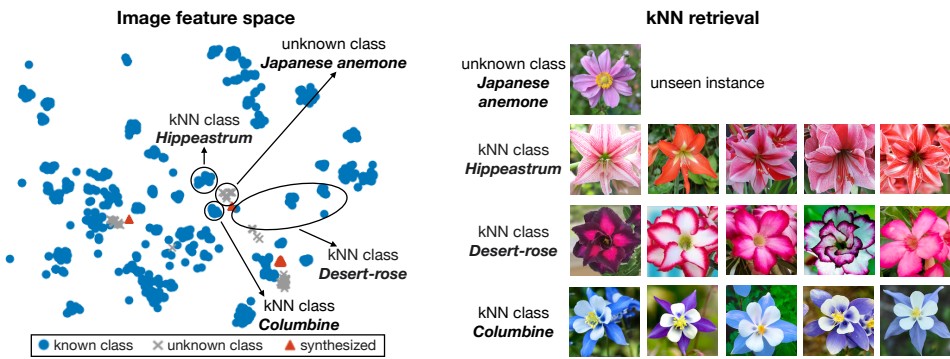

Figure 3: Visualizing image feature synthesis based on the **joint extrapolation** scheme (Eq. (5)) on Flowers102 dataset. Note our feature generator is not trained on the unknown classes, but can still synthesize faithful image features (red triangle) lying close to the real ones (gray cross). This is achieved by extrapolating an unseen instance from the kNN class examples (only a random one per kNN class is used), effectively combining their related patterns like the shape and texture of flowers.

dynamics that could also impact the ID-OOD performance tradeoff, especially with long finetuning runs. Take Fig. 1(b) for example. Without proper regularization, the CoOp baseline achieves either suboptimal ID performance at early epochs, or saturated ID performance but decreasing OOD performance (*i.e.*, overfitting) later on. To address this issue, we introduce an adaptive self-distillation method that regularizes optimization dynamics to further reduce overfitting.

More specifically, we use the model checkpoints from earlier epochs (*i.e.*, teacher model often with less overfitting) to guide optimization at the current epoch (*i.e.*, student model often with more overfitting). Since the CLIP model is frozen during prompt learning, the "model" we consider here is our feature generator $\theta$ whose synthesized OOD features will impact the joint ID-OOD optimization. Hence we enforce the consistency between the final prediction probabilities (Eq. (4) or (6)) induced by the teacher model $p^T$ and student model $p^S$ using the mean squared error $\text{MSE}(p^T, p^S)$. Ideally, this will help us to avoid OOD performance drop while preserving the ID performance.

The key to our self-distillation method is the choice of teacher model $\theta^T$. Obviously, selecting $\theta^T$ as one single model checkpoint at a historical epoch time is unlikely to strike a good trade-off between the ID and OOD performance. Mean Teacher (MT) (Tarvainen & Valpola, 2017) is a better alternative, which calculates an Exponential Moving Average (EMA) over the past checkpoints up until the current time $t$ (Eq. (7)). Here we propose Adaptive Local Mean Teacher (ALMT) that extends MT in two ways: 1) calculating EMA only within a local time window $[t - m_t, t]$ using the last $m_t$ checkpoints. This avoids the negative impact on the teacher's ID performance from those underfit early checkpoints. 2) the window size $m_t$ is time-adaptive such that $m_t$ is small in the early stage of finetuning (for the same purpose of ruling out underfit checkpoints), and then $m_t$ gradually increases in order to cover older checkpoints with improved ID performance but less overfitting. Such curriculum is summarized in Eq. (8) as below:

$$\textbf{MT}_{[1,t]}: \quad \theta_i^T = \alpha \theta_{i-1}^T + (1 - \alpha)\theta_i, \quad for \quad i = \{1, \ldots, t\}, \tag{7}$$

$$\textbf{ALMT}_t: \quad \textbf{MT}_{[t-m_t,t]}, \quad m_t = \left\lfloor \left(1 + \cos\left(\frac{t_{\max} + t}{t_{\max}}\pi\right)\right) \cdot \frac{1}{2}(m_{\max} - m_{\min}) + m_{\min} \right\rfloor, \tag{8}$$

where $m_{\max} = 9, m_{\min} = 2$, $t_{\max}$ is the maximum number of finetuning epochs, and the window size $m_t$ is increased following a cosine schedule. Note our ALMT method requires maintaining a queue of past $m_t$ checkpoints and re-calculating EMA for each time $t$, both of which are cheap thanks to our compact model size of $\theta_i$ and the small window size $m_t \in \{2, \ldots, 9\}$.

## 4 EXPERIMENTS

We evaluate OOD generalization under the two settings introduced in (Zhou et al., 2022b) (more details in Appendix A): 1) generalization from ID (base) to OOD (new) classes within one dataset. The base and new class splits are used for finetuning and evaluation respectively. 2) cross-dataset generalization with one ID dataset for finetuning and other datasets for OOD evaluation. The cross-

Table 1: **Base-to-new class generalization**. Our OGEN approach consistently improves the new class generalization for all prompt learning baselines on average (across 11 datasets). OGEN also maintains or improves the average performance on base classes. H: Harmonic mean of base and new accuracies (%).

| | +OGEN | CoOp ✗ | CoOp ✓ | CoCoOp ✗ | CoCoOp ✓ | VPT ✗ | VPT ✓ | SHIP ✗ | SHIP ✓ | KgCoOp ✗ | KgCoOp ✓ | MaPLe ✗ | MaPLe ✓ | PromptSRC ✗ | PromptSRC ✓ |
|---|---|---|---|---|---|---|---|---|---|---|---|---|---|---|---|
| Avg across 11 datasets | Base | 82.69 | **83.47** | **80.47** | 79.86 | 82.51 | **82.52** | 80.03 | **80.79** | 80.73 | **81.34** | 82.28 | **82.40** | **84.26** | 84.17 |
| | New | 63.22 | **69.54** | 71.69 | **73.35** | 69.01 | **70.61** | 73.69 | **76.14** | 73.60 | **75.68** | 75.14 | **76.37** | 76.10 | **76.86** |
| | Δ | | **+6.32** | | **+1.66** | | **+1.60** | | **+2.45** | | **+2.08** | | **+1.23** | | **+0.76** |
| | H | 71.66 | **75.87** | 75.83 | **76.47** | 75.16 | **76.10** | 76.73 | **78.40** | 77.00 | **78.40** | 78.55 | **79.27** | 79.97 | **80.34** |
| ImageNet | Base | 76.47 | 76.40 | 75.98 | 76.50 | 75.96 | 75.09 | 75.87 | 76.14 | 75.83 | 75.88 | 76.66 | 77.02 | 77.60 | 77.50 |
| | New | 67.88 | 68.80 | 70.43 | 70.23 | 67.32 | 67.66 | 69.95 | 71.18 | 69.96 | 70.93 | 70.54 | 70.73 | 70.73 | 70.97 |
| | H | 71.92 | 72.40 | 73.10 | 73.23 | 71.38 | 71.18 | 72.79 | 73.58 | 72.78 | 73.32 | 73.47 | 73.74 | 74.01 | 74.09 |
| Caltech101 | Base | 98.00 | 96.67 | 97.96 | 96.67 | 97.50 | 96.33 | 97.55 | 98.09 | 97.72 | 98.52 | 97.74 | 98.37 | 98.10 | 98.32 |
| | New | 89.81 | 92.61 | 93.81 | 94.79 | 94.10 | 92.36 | 95.20 | 95.26 | 94.39 | 94.12 | 94.36 | 94.54 | 94.03 | 94.76 |
| | H | 93.73 | 94.59 | 95.84 | 95.72 | 95.77 | 94.30 | 96.36 | 96.65 | 96.03 | 96.27 | 96.02 | 96.42 | 96.02 | 96.50 |
| OxfordPets | Base | 93.67 | 95.18 | 95.20 | 96.49 | 96.05 | 96.05 | 95.37 | 96.95 | 94.65 | 95.91 | 95.43 | 95.11 | 95.33 | 95.96 |
| | New | 95.29 | 96.45 | 97.69 | 97.86 | 95.84 | 96.84 | 97.87 | 97.33 | 97.76 | 97.65 | 97.76 | 97.89 | 97.30 | 97.48 |
| | H | 94.47 | 95.81 | 96.43 | 97.17 | 95.94 | 96.45 | 96.61 | 97.14 | 96.18 | 96.77 | 96.58 | 96.47 | 96.30 | 96.71 |
| Stanford Cars | Base | 78.12 | 78.65 | 70.49 | 68.96 | 75.00 | 74.23 | 68.57 | 68.63 | 71.76 | 71.86 | 72.94 | 73.63 | 78.27 | 77.59 |
| | New | 60.40 | 65.28 | 73.59 | 74.23 | 63.45 | 67.97 | 73.90 | 75.45 | 75.04 | 75.95 | 74.00 | 74.30 | 74.97 | 75.17 |
| | H | 68.13 | 71.35 | 72.01 | 71.50 | 68.74 | 70.96 | 71.14 | 71.88 | 73.36 | 73.84 | 73.47 | 73.96 | 76.58 | 76.38 |
| Flowers102 | Base | 97.60 | 97.38 | 94.87 | 93.95 | 96.89 | 98.03 | 94.02 | 94.67 | 95.00 | 95.83 | 95.92 | 96.52 | 98.07 | 97.34 |
| | New | 59.67 | 67.70 | 71.75 | 72.08 | 70.02 | 69.15 | 74.40 | 76.49 | 74.73 | 74.75 | 72.46 | 74.46 | 76.50 | 77.67 |
| | H | 74.06 | 79.87 | 81.71 | 81.57 | 81.29 | 81.09 | 83.06 | 84.61 | 83.65 | 83.98 | 82.56 | 84.06 | 85.95 | 86.39 |
| Food101 | Base | 88.33 | 89.21 | 90.70 | 91.17 | 88.88 | 91.50 | 90.54 | 91.07 | 90.50 | 90.80 | 90.71 | 91.02 | 90.67 | 90.69 |
| | New | 82.26 | 87.22 | 91.29 | 91.67 | 88.95 | 88.53 | 91.03 | 92.79 | 91.70 | 92.01 | 92.05 | 92.02 | 91.53 | 91.68 |
| | H | 85.19 | 88.21 | 90.99 | 91.42 | 88.91 | 89.99 | 90.78 | 91.92 | 91.09 | 91.40 | 91.38 | 91.52 | 91.10 | 91.19 |
| FGVC Aircraft | Base | 40.44 | 41.67 | 33.41 | 35.33 | 38.33 | 39.33 | 34.27 | 35.47 | 36.21 | 37.08 | 37.44 | 37.07 | 42.73 | 41.26 |
| | New | 22.30 | 29.14 | 23.71 | 34.41 | 25.27 | 26.55 | 32.33 | 34.32 | 33.55 | 37.19 | 35.61 | 37.41 | 37.87 | 40.26 |
| | H | 28.75 | 34.29 | 27.74 | 34.86 | 30.46 | 31.70 | 33.28 | 34.89 | 34.83 | 37.14 | 36.50 | 37.24 | 40.15 | 40.75 |
| SUN397 | Base | 80.60 | 80.86 | 79.74 | 80.27 | 80.27 | 79.06 | 79.54 | 81.14 | 80.29 | 81.91 | 80.82 | 81.06 | 82.67 | 82.57 |
| | New | 65.89 | 67.49 | 76.86 | 75.69 | 74.36 | 74.49 | 75.27 | 75.94 | 76.53 | 78.83 | 78.70 | 81.07 | 78.47 | 78.83 |
| | H | 72.51 | 73.57 | 78.27 | 77.91 | 77.20 | 76.71 | 77.35 | 78.45 | 78.36 | 80.34 | 79.75 | 81.06 | 80.52 | 80.65 |
| DTD | Base | 79.44 | 79.16 | 77.01 | 75.00 | 77.08 | 77.43 | 74.88 | 76.02 | 77.55 | 78.01 | 80.36 | 79.73 | 83.37 | 83.75 |
| | New | 41.18 | 50.96 | 56.00 | 56.44 | 53.62 | 55.79 | 56.88 | 64.62 | 58.64 | 62.56 | 59.18 | 62.68 | 62.97 | 62.54 |
| | H | 54.24 | 62.01 | 64.85 | 64.41 | 63.24 | 64.85 | 64.65 | 69.86 | 64.35 | 69.43 | 68.16 | 70.18 | 71.75 | 71.60 |
| EuroSAT | Base | 92.19 | 91.67 | 87.49 | 78.33 | 91.67 | 90.00 | 88.62 | 89.17 | 85.64 | 86.05 | 94.07 | 93.83 | 92.90 | 93.40 |
| | New | 54.74 | 73.51 | 60.04 | 64.69 | 58.31 | 66.75 | 66.87 | 74.28 | 64.34 | 70.18 | 73.23 | 74.30 | 73.90 | 76.74 |
| | H | 68.69 | 81.59 | 71.21 | 70.86 | 71.28 | 76.65 | 76.22 | 81.05 | 73.48 | 77.30 | 82.35 | 82.93 | 82.32 | 84.25 |
| UCF101 | Base | 84.69 | 91.33 | 82.33 | 85.78 | 80.07 | 90.68 | 81.08 | 81.33 | 82.89 | 82.84 | 83.00 | 82.99 | 87.10 | 87.44 |
| | New | 56.05 | 65.81 | 73.45 | 74.78 | 74.50 | 70.54 | 76.85 | 79.83 | 76.67 | 78.28 | 78.66 | 80.68 | 78.80 | 79.28 |
| | H | 67.46 | 76.50 | 77.64 | 79.90 | 77.18 | 79.35 | 78.91 | 80.57 | 79.65 | 80.49 | 80.77 | 81.82 | 82.74 | 83.16 |

dataset setting is more challenging since there will be both domain- and class-incremental distribution shift, *e.g.*, from generic object classification on ImageNet (Deng et al., 2009) to satellite imagery recognition on EuroSAT (Helber et al., 2019).

**Datasets.** For both settings we use 11 datasets: ImageNet (Deng et al., 2009), Caltech101 (Fei-Fei et al., 2004), OxfordPets (Parkhi et al., 2012), StanfordCars (Krause et al., 2013), Flowers102 (Nilsback & Zisserman, 2008), Food101 (Bossard et al., 2014), FGVC-Aircraft (Maji et al., 2013), SUN397 (Xiao et al., 2010), UCF101 (Soomro et al., 2012), DTD (Cimpoi et al., 2014) and EuroSAT (Helber et al., 2019).

**Baselines.** For finetuning, we consider prompt learning approaches CoOp (Zhou et al., 2022a), CoCoOp (Zhou et al., 2022b), VPT (Jia et al., 2022), and the state-of-the-art methods SHIP (Wang et al., 2023), KgCoOp (Yao et al., 2023), MaPLe (Khattak et al., 2023a) and PromptSRC (Khattak et al., 2023b). For each baseline, we apply our method (dubbed OGEN) to obtain an OOD GENeralization improved version. For fairness, we use the same implementation details of each baseline, including the prompt length, vision backbone in CLIP (Radford et al., 2021) (*i.e.*, ViT-B/16) and train/test data splitting. The reported results are an average over three random seeds.

## 4.1 GENERALIZATION FROM BASE TO NEW CLASSES

The base-to-new generalization setting creates a strictly class-incremental distribution shift since the base and new class splits in one dataset are disjoint. All prompt learners are trained on the base classes, and tested on the base and new classes separately to evaluate the trade-off between ID and OOD performance. Here we follow (Xian et al., 2017) to report the harmonic mean of base and new class accuracies to quantify such trade-off.

Table 2: **Cross-dataset generalization**: CLIP finetuning (prompt learning) on the source dataset ImageNet, followed by testing on 10 target datasets. Our method OGEN improves the generalization performance of both CoOp and CoCoOp on all the target datasets.

| | Source | Target | | | | | | | | | | |
|---|---|---|---|---|---|---|---|---|---|---|---|---|
| | ImageNet | Caltech101 | OxfordPets | StanfordCars | Flowers102 | Food101 | FGVCAir | SUN397 | DTD | EuroSAT | UCF101 | *Average* |
| CoOp | 71.51 | 93.70 | 89.14 | 64.51 | 68.71 | 85.30 | 18.47 | 64.15 | 41.92 | 46.39 | 66.55 | 63.88 |
| OGEN-CoOp | **71.52** | 94.60 | 90.73 | 65.07 | 70.55 | 87.26 | 19.84 | 65.77 | 44.90 | 49.53 | 69.36 | **65.76** |
| CoCoOp | 71.02 | 94.43 | 90.14 | 65.32 | 71.88 | 86.06 | 22.94 | 67.36 | 45.73 | 45.37 | 68.21 | 65.74 |
| OGEN-CoCoOp | **71.28** | 95.12 | 91.37 | 66.04 | 72.90 | 86.54 | 22.95 | 68.42 | 46.38 | 45.82 | 69.74 | **66.53** |

Table 1 summarizes the results on 11 datasets. On average, our OGEN method consistently improves the new class accuracy for all the prompt learning baselines. CoOp is particularly interesting since its default learning schedule (200 epochs) is much longer than that of CoCoOp and VPT (10 epochs). Without proper regularization, CoOp inevitably shows more serious overfitting to the base classes (82.69% on average) with low performance on new classes (63.22%) after long training runs. Our OGEN is especially useful in this case, significantly improving the average new class accuracy of CoOp from 63.22% to 69.54%. As also visualized in Appendix C - Fig. 6(a), the new class generalization sees notable gains on 3 datasets — DTD for texture classification, EuroSAT for satellite image recognition and UCF101 for action recognition, which all demonstrate large inter-class variations. This validates the superior generalizability of OGEN, thanks to its capability of OOD feature synthesis and regularization. OGEN also improves the average base class accuracy of CoOp from 82.69% to 83.47%. Specifically, OGEN improves on 6 datasets with negligible performance drop on other 5, see Fig. 6(b). The gains on base classes can be attributed to 1) the joint discrimination of known and unknown classes and 2) our adaptive self-distillation method that strikes a good ID-OOD performance tradeoff.

For CoCoOp and VPT with a significantly shorter training schedule, they suffer from much less overfitting with higher new but lower base accuracies than CoOp. This makes our OGEN unable to unleash its full potential to address overfitting. That said, we find both OGEN-CoCoOp and OGEN-VPT can still improve the average new class accuracy while achieving a similar base class accuracy. We are likely to further improve the base accuracy when given a longer optimization schedule that allows more ID-OOD performance balancing.

Among the state-of-the-art methods, SHIP (+CoOp) and PromptSRC are related to our OGEN approach in the use of similar techniques of feature synthesis and self-regularization respectively. Table 1 shows that OGEN can improve the new class generalization of both SHIP and PromptSRC by exploring the synergy between regularization and OOD feature synthesis. OGEN also consistently improves the average base and new class accuracies for KgCoOp and MaPLe. Fig. 5 uses KgCoOp to exemplify how these methods still suffer from overfitting (although reduced to some extent by various techniques), and how our OGEN improves the learning curves of both base and new classes. It is worth noting that different methods are trained for different numbers of epochs, thus again, they have different levels of overfitting. OGEN improves generalization more over SHIP (200 epochs) and KgCoOp (100 epochs) with long learning schedules (more serious overfitting). Our gains are smaller over MaPLe (5 epochs) and PromptSRC (20 epochs) with short training runs, but larger gains are expected when trained for longer runs.

## 4.2 CROSS-DATASET GENERALIZATION

Table 2 shows the generalization performance from ImageNet to 10 target datasets. We consider the representative CoOp and CoCoOp baselines with long and short training runs respectively. As shown in the table, our OGEN uniformly improves the generalization performance (across baselines and target datasets) with competitive source dataset performance. The improvements are especially large on those low performing datasets DTD, EuroSAT, UCF101 with large distribution shift from ImageNet. This highlights the benefits of our OOD feature generation module. OGEN also obtains reasonable gains on the high performing datasets like OxfordPets that contains similar classes (*e.g.*, different dog breeds) with ImageNet, demonstrating the universality of our approach.

Table 3: Ablating our **class-conditional feature generator** $\theta$ (Eq. (5)) and **self-distillation** method ALMT (Eq. (8)). H: Harmonic mean of the base and new class accuracies averaged on 11 datasets.

| $\theta$ | ALMT | Base | New | H |
|---|---|---|---|---|
| ✗ | ✗ | 82.69±1.08 | 63.22±0.51 | 71.66±0.54 |
| ✓ | ✗ | 82.49±0.95 | 69.02±0.64 | 75.15±1.12 |
| ✓ | ✓ | **83.47**±0.30 | **69.54**±0.34 | **75.88**±0.11 |

Table 4: **Class-conditional feature generator**: different design choices of no extrapolation from kNN classes, extrapolating per class (Eq. (3)) and extrapolating jointly (Eq. (5)).

| | Base | New | H |
|---|---|---|---|
| No Extrap | **83.34**±0.26 | 64.08±0.95 | 72.46±0.68 |
| Extrap per class | 82.90±0.35 | 66.04±0.89 | 73.52±0.63 |
| Extrap jointly | 82.49±0.95 | **69.02**±1.25 | **75.15**±1.12 |

Table 5: **Class-conditional feature generator**: kNN retrieval vs. Random sampling of known classes with varying $K$.

| | $K$ | Base | New | H |
|---|---|---|---|---|
| kNN | 1 | **82.76**±0.49 | 67.01±1.35 | 74.06±0.63 |
| | 2 | 82.35±0.76 | 67.79±2.37 | 74.36±1.19 |
| | 3 | 82.49±0.95 | **69.02**±1.25 | **75.15**±1.12 |
| | 4 | 82.37±0.46 | 68.85±0.52 | 75.00±0.13 |
| Rand | 3 | 81.69±0.35 | 68.30±0.38 | 74.40±0.36 |

Table 6: **Self-distillation**: Mean Teacher (MT) vs. Adaptive Local Mean Teacher (ALMT) with fixed window size $m$ or adaptive $m_t$ (default).

| | Base | New | H |
|---|---|---|---|
| No distillation | 82.49±0.65 | 69.02±0.64 | 75.15±1.12 |
| MT | 83.34±0.15 | 68.30±0.77 | 75.08±0.47 |
| ALMT ($m = 2$) | 81.70±0.59 | 68.47±0.59 | 74.50±0.55 |
| ALMT ($m = 9$) | 82.21±0.80 | 68.57±0.85 | 74.77±0.29 |
| ALMT ($m_t$) | **83.47**±0.30 | **69.54**±0.34 | **75.88**±0.11 |

## 4.3 ABLATION STUDIES

Our ablation studies are conducted using OGEN-CoOp with a meaningfully long learning schedule. We start with ablating the two main components of OGEN: class-conditional feature generator $\theta$ and adaptive self-distillation method ALMT. Table 3 confirms both components are useful (more visuals in Appendix B). We see the feature generator improves the new class accuracy by a large margin without hurting the base class accuracy. This suggests the high quality of its generated OOD features and the need of joint ID-OOD feature optimization. ALMT is shown to further improve on both base and new classes, leading to a higher Harmonic mean and a much lower performance variance. This highlights the need of regularizing joint optimization for a good performance tradeoff.

Table 4 compares different design choices of our class-conditional feature generator. Recall that we adopt an extrapolation scheme that extrapolates new image features from the kNN base class features. What if we use no extrapolation at all, and directly learn a mapping from the new class name to new image features? As shown in the table, this only slightly helps the new class generalization probably because the generated features are not faithful enough from an unconstrained text-to-image mapping. Then between the "Extrapolating per class" and "Extrapolating jointly" schemes, the latter improves on new classes much more significantly, showing the benefits of collaborative class relation modeling for extrapolation. Table 5 further ablates on the number of kNN base classes needed for extrapolation, arriving at the best $K = 3$. By comparison, randomly selecting 3 base classes does not perform as well. Finally, Appendix D illustrates the advantage of our feature synthesis approach over replay-based methods using real data.

Table 6 compares various self-distillation baselines applied on top of our feature generator. Notably, the simple Mean Teacher (MT) is not helping, which inspires us to use a local version to completely rule out the early-stage underfit model checkpoints. We further propose Adaptive Local Mean Teacher (ALMT) that calculates EMA within a local time window of increasing size $m_t$ (from $m_{\min} = 2$ to $m_{\max} = 9$). As shown in the table, ALMT achieves the best performance due to the adaptiveness of $m_t$, which effectively avoids both the underfitting (from early epochs) and overfitting (from recent epochs) effects in the teacher model. Apparently, this is not possible with a fixed window size ($m = m_{\min}$ or $m_{\max}$) which hurts performance.

## 5 CONCLUSIONS AND FUTURE WORK

In this paper, we study the OOD generalization of recent CLIP finetuners and propose an effective approach to reduce their overfitting to seen classes. For that, a class-conditional feature generator is used to synthesize OOD features for joint optimization, and the optimization dynamics are further regularized by an adaptive distillation scheme. The superior generalization capability of our approach is demonstrated under different OOD settings. In the future, we plan to go beyond prompt learning and evaluate how much our benefits hold for other finetuning methods like adaptor tuning. Moreover, it would be interesting to figure out how well our "unknown-aware" approach can model uncertainties on unseen data, which can be evaluated on existing OOD detection benchmarks.

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

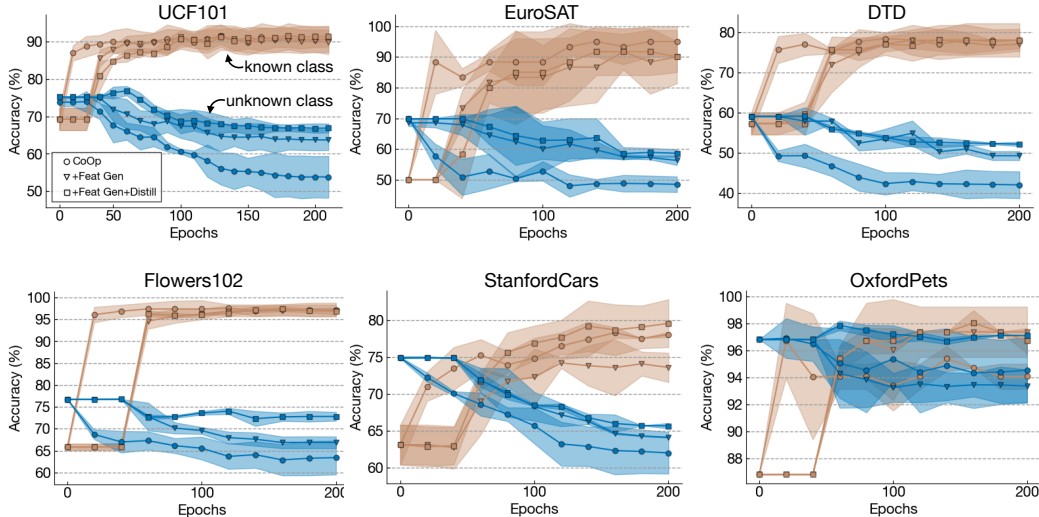

Figure 4: More example learning curves of the long finetuning runs (200 epochs) with CoOp (Zhou et al., 2022a) method. Under the within-dataset generalization setting, CoOp typically overfits the known classes and achieves decreasing accuracy for the unknown classes. The class-conditional feature generator plays a key role in our full method OGEN, which reduces overfitting by generating OOD features for the unknown-aware optimization. Our adaptive self-distillation method further reduces overfitting via regularizing the optimization dynamics.

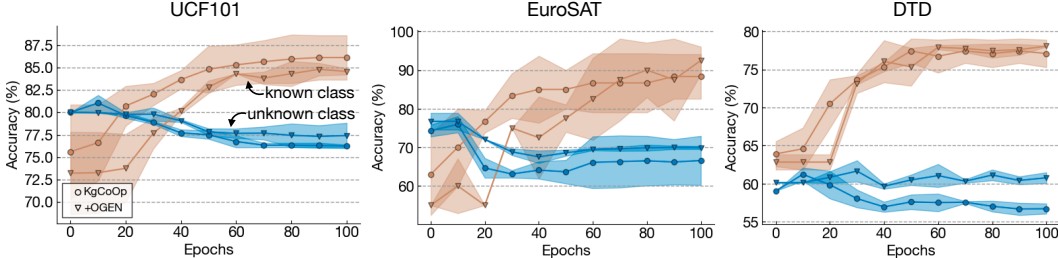

Figure 5: Learning curves of the long finetuning runs (100 epochs) with KgCoOp (Yao et al., 2023) vs. OGEN-KgCoOp methods (within-dataset generalization setting). Despite the overfitting reducing technique used in KgCoOp, it still suffers from some extent of overfitting. See how OGEN often improves the learning curves of both base and new classes.

## A    REMARKS ON PROBLEM DEFINITION

The focus of this paper is to understand the behaviors of and improve OOD generalization of CLIP finetuning. As shown in Fig. 1(a) of the main paper, the CLIP model can be finetuned on various downstream tasks before evaluating OOD generalization. Two settings are considered: 1) within-dataset generalization where one dataset has ID (base) vs. OOD (new) class splits for finetuning and evaluation respectively; 2) cross-dataset generalization with one ID dataset for finetuning and other OOD datasets for evaluation.

Note CLIP is pretrained on enormous volumes of data, which inevitably could have class overlap with some "OOD" data for evaluation. Also, there could be potential class overlap between the curated ID and OOD data themselves, *e.g.*, under the cross-dataset generalization setting where the dataset pair may include similar categories in their class taxonomies. Therefore, we consider more of a generalized OOD generalization test for the large-scale pretrained CLIP model. That is, whenever the class overlap happens, the domain-incremental distribution shift is considered in our evaluation, otherwise we evaluate under the strictly class-incremental distribution shift.

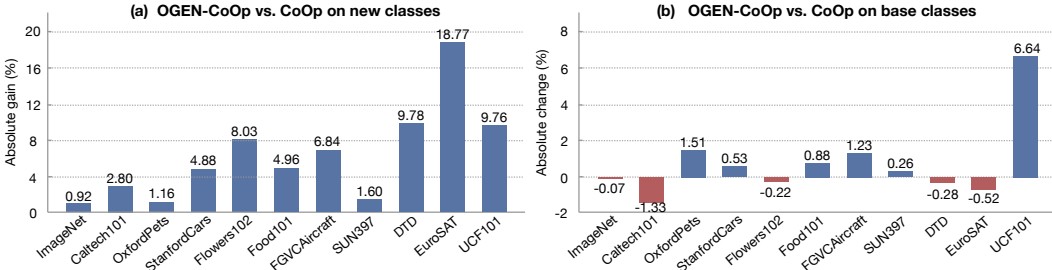

Figure 6: **Base-to-new class generalization** when our OGEN approach is applied to the CoOp baseline that suffers from overfitting due to a long learning schedule (200 epochs). OGEN largely overcomes overfitting and **(a)** improves OOD generalization on new classes for all 11 datasets, sometimes by a large margin. **(b)** At the same time, OGEN is able to improve the base class accuracies on most datasets, with only minor accuracy drop on a few others.

## B    OVERFITTING WITH CLIP FINETUNING: MORE EXAMPLES

Fig. 4 shows more learning curves of the prompt learning method CoOp (Zhou et al., 2022a) that finetunes CLIP on a long schedule. Clearly, CoOp overfits on all the considered datasets with decreasing generalization performance on the unknown classes. While both components of our proposed OGEN method – class-conditional feature generator and adaptive self-distillation – are found useful to address overfitting. Our feature generator is observed to play a key role by generating OOD features for the following unknown-aware optimization and regularization.

Fig. 5 shows example learning curves of one of the state-of-the-art methods KgCoOp (Yao et al., 2023). This method has relatively long finetuning runs by default (100 epochs). We can see that KgCoOp still suffers from some extent of overfitting, although there is an overfitting reducing component in it. Our OGEN can further alleviate overfitting with KgCoOp, improving its learning curves of both base and new classes.

## C    VISUALIZING PER-DATASET RESULTS

Fig. 6 breaks down the performance gap between CoOp and our OGEN-CoOp on both the base and new classes for 11 datasets. The base-to-new class generalization setting is considered on each dataset.

## D    FEATURE SYNTHESIS VS. REPLAY-BASED METHOD

Recall the goal of our class-conditional feature generator is to model the vast space and complex distribution of unknown class data in OOD domains. Here we investigate how well our synthesized OOD features can represent the real world of unknown data. Specifically, we explore the use of replay methods by sampling real OOD data from the large-scale LAION-400M dataset (Schuhmann et al., 2021) and using the sampled data for replay. We compare our synthesized OOD features against those real OOD data in terms of their contribution to training regularization on downstream tasks.

**Experimental details.** We experiment under the base-to-new class generalization setting where CoOp (Zhou et al., 2022a) is the CLIP finetuning baseline on each of the 11 downstream datasets. For fair comparison between the OGEN- and replay-boosted CoOp variants, we always use ALMT with the only difference in the OOD data source (synthesis vs. replay).

**Sampling of replay data.**

- Class filtering: to ensure the replay data serve as a good proxy of OOD data for each downstream task, we need to perform class filtering when sampling image-text-pairs from LAION-400M, *i.e.*, we filter image-text-pairs that are semantically close to those "known" classes on the considered dataset. We do so by using CLIP to calculate the cosine similarity

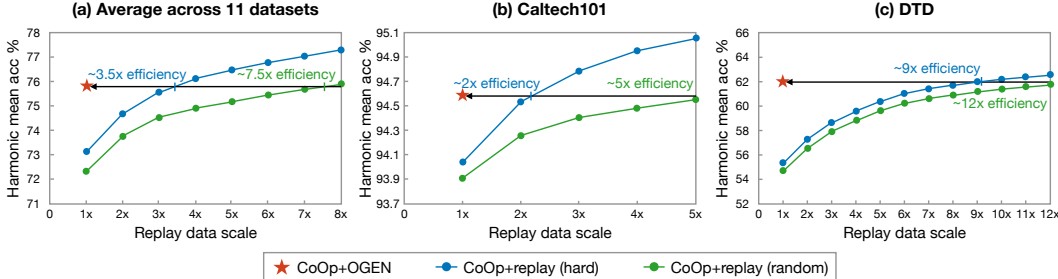

Figure 7: **OOD feature synthesis is much more data efficient than replaying real OOD data.** Real OOD data are sampled from LAION-400M dataset with varying data scale, *i.e.*, multiple times more than the synthesized data. Here we use the CoOp baseline for base-to-new class generalization, and measure the Harmonic mean of base and new accuracies. **(a)** We can see that OGEN with OOD feature synthesis creates about 3.5x and 7.5x gain in data efficiency on average across 11 datasets, when compared to the replay method with hard sample mining and random sampling strategy respectively (see text for details). **(b-c)** The replayed OOD samples, despite being data inefficient, are more helpful when they have a closer semantic or data distribution with the downstream dataset (*e.g.*, Caltech101) so to act as semi-hard negatives. They are far less useful for distributionally different DTD, the texture database. On the other hand, OGEN benefits from dataset-specific OOD feature synthesis, which often offers hard negatives to boost data efficiency for training.

between the text features of the query text and known class names, and then dropping query texts with the maximum similarity score higher than 0.82. Note our synthesized features are guaranteed to be OOD since they are generated in a class-conditional manner with the query classes disjoint from known classes.

- Sampling strategy: with class filtering in place, we consider both random data sampling and hard sample mining to retrieve replay data from LAION-400M. For hard sample mining, we first rank the query texts from a sufficiently large pool (randomly sampled) in terms of the maximum similarity score mentioned above. Then we simply select the top similar query texts and use the corresponding images as hard replay data (similar to hard negatives from hard negative sampling).

- Sampling data size: given the large amount of image-text-pairs in LAION-400M, we increase the data size of the replay data (either random or hard) by 1-to-12x more than the size of our synthesized OOD data on each downstream dataset. Note our feature generator synthesizes the same amount of OOD features as the "unknown" class splits of the training set of considered dataset.

**Observations from Fig. 7.**

- Better performance is generally obtained with the use of more real OOD data for replay, and performance grows faster with hard mined replay data. When sufficient replay data are used, the performance can surpass that of our synthesized OOD features. This demonstrates the benefits of sampling big and diverse OOD data for training regularization purposes.

- However, our feature synthesis approach is much more data efficient than replay-based methods. The key reason behind such advantage is that our feature generator is more likely to generate hard OOD samples to better regularize decision boundaries, in comparison to real-world samples. In our case, 1) the feature generator itself is trained on the downstream dataset, thus can synthesize dataset-specific OOD features that adapt better to the task at hand. 2) Recall that we extrapolate OOD features from kNN known class features. This suggests there is inevitable shared information between the known class and synthesized features, further increasing the hardness of the latter. On the other hand, both of the aforementioned factors are missing for the real OOD data sampled from a separate domain, which contributes to their data inefficiency in training regularization. Real OOD data are most useful when they are distributionally similar to the downstream dataset (*e.g.*, Caltech101) and hence can act as semi-hard negatives. Otherwise, data efficiency will see a significant drop when replaying OOD data on distributionally distant dataset like DTD. On

average (across 11 datasets), our feature synthesis method is about 3.5x and 7.5x more data efficient than the replay method with hard mining and random sampling respectively.

**Summary.** Replay-based methods perform well with large data size, but suffer from low data efficiency as well as large memory cost (to maintain replay data). Our class-conditional feature generator avoids these issues by synthesizing hard OOD features on the fly. Note our feature generator is lightweight and only incurs small computational cost. Its runtime on GPU is 0.019 seconds, which is significantly smaller than that of the feature extraction step of CLIP (text encoder: 0.046 seconds, image encoder: 1.016 seconds). One promising future direction is the combination of our feature synthesis method and replay methods, aiming to take advantage of their respective benefits of data efficiency and diversity.

