# OpenReview forum: "Overcoming the Pitfalls of Vision-Language Model Finetuning for OOD Generalization"
_ICLR.cc/2024/Conference — ICLR 2024 poster_

### Official Review · Reviewer_v4h7 · 2023-10-22

**Soundness:** 3 good
**Presentation:** 3 good
**Contribution:** 3 good
**Rating:** 5
**Confidence:** 5

**Summary:**

The paper works on improving the OOD generalization of finetuned vision-language models. Specifically, a class-conditional feature generator and adaptive self-distillation mechanism are proposed to serve the goal.

**Strengths:**

[$\textbf{Interesting Idea}$] The idea of generating unknown-class features is interesting.

[$\textbf{Presentation Quality}$] The presentation is clear and easy to follow.

**Weaknesses:**

[$\textbf{Unconvincing Statement}$] The work mentions that it is the first to unveil the pitfalls of finetuning VLMs by prompt learning can cause overfitting on base classes, resulting in poor performance on novel classes. However, CoCoOp “Conditional Prompt Learning for Vision-Language Models, CVPR 2022” has already observed this and proposed conditional prompt learning to address it.

[$\textbf{Unclear Model Design}$] This work uses known image and text features as K and V, while unknown text features as Q to generate unknown image features. The rationale behind this is not clear. It would be nice to explain this in more details.

[$\textbf{Missing Related Works}$] For finetuning methods, there are many works that need to be discussed, e.g., “CLIP-Adapter: Better Vision-Language Models with Feature Adapters”, “Task Residual for Tuning Vision-Language Models”, “Improving Zero-Shot Generalization for CLIP with Synthesized Prompts”, “MaPLe: Multi-modal Prompt Learning” and “Self-regulating Prompts: Foundational Model Adaptation without Forgetting”.

[$\textbf{Small Performance Gains}$] The results in Table 1 show the improvements from adding the proposed method are rather limited. Moreover, the performance is much worse than some SOTA methods, e.g., “MaPLe: Multi-modal Prompt Learning, CVPR 2023”, “Improving Zero-Shot Generalization for CLIP with Synthesized Prompts, ICCV 2023” and “Self-regulating Prompts: Foundational Model Adaptation without Forgetting, ICCV 2023”. It would be nice to see the performance of these SOTA methods by adding the proposed components.

**Questions:**

Please refer to the weaknesses.

---

> ### Author Response · Authors · 2023-11-20
> **Response to Reviewer v4h7**
>
> Thanks for the detailed feedback. Please find below our response to each comment.
>
> **Unconvincing statement: CoCoOp already observes overfitting with a method proposed to address it.**
>
> We humbly argue that we are the first to provide a ``comprehensive’’ study of overfitting across different finetuning baselines, datasets and OOD settings. Hence in the revised paper, we will tone down the point about being the first and instead focus on our contribution of being comprehensive. Algorithm-wise, instead of using input-conditional prompts in CoCoOp that implicitly tackles overfitting, we propose OGEN to directly reduce overfitting via unknown-aware optimization with OOD feature synthesis and adaptive model distillation. Our benefits are validated by the gains when applying OGEN on top of CoCoOp (Table 1 & 2).
>
> **Unclear model design: explain the rationale behind the attention method used to generate unknown image features.**
>
> Please refer to Section 3.2 (3rd paragraph) for the reasoning. At high level, to generate unknown image features from just the class name (using its text features), we found it's hard to directly learn the text-to-image feature mapping. This is backed by our ablations in Table 4 - see the poorly performing ``no extrapolation'' baseline. Then we choose to condition the feature generation process on some knowledge about known classes (i.e. the image/text features of kNN known classes), arriving at an extrapolation solution which is easier. The remainder of Section 3.2 introduces two extrapolation schemes, both naturally implemented by cross-attention. The attention mechanism is similar to measuring similarities between the unknown text (query) and kNN known text (key) features first, and then using such text similarities to extrapolate unknown image features from kNN known image features (value). We will clarify more on such rationale in the revised paper.
>
> **Missing related works.**
>
> We will add discussions and empirical comparisons with the suggested works in the revised paper. Comparing results for many of the new works can also be found in our “Response to common concern: comparing with the SOTA”.
>
> **Small performance gains in Table 1, and results of OGEN+SOTA methods.**
>
> Our observations from Table 1 are that 1) OGEN achieves notable improvements on new classes, 2) while its base class accuracy remains competitive. This is the desired behavior to us since we focus more on the new class generalization, although this may make the gains in the Harmonic mean of base and new accuracies seem smaller. Table 2 is another example where OGEN achieves convincing gains on new datasets without hurting the source ImageNet performance. For SOTA results, please refer to our “Response to common concern: comparing with the SOTA”.

---

> > ### Author Response · Authors · 2023-11-22
> > **Gentle Reminder of the Discussion Deadline**
> >
> > Dear Reviewer v4h7, thanks again for your insightful feedback! Taking into account your tight schedule, we are sending you this gentle reminder that the rebuttal deadline is approaching. Would you mind letting us know if your concerns have been fully addressed or if you have more questions? In the latter case, we will try our best to address any lingering questions before the deadline. Looking forward to your reply! In the meantime, we will be continuously improving the paper following your suggestions and working towards code release. Thanks for your time!

---

### Official Review · Reviewer_Rp4n · 2023-10-25

**Soundness:** 3 good
**Presentation:** 3 good
**Contribution:** 3 good
**Rating:** 8
**Confidence:** 4

**Summary:**

This paper proposes a method to regularize ERM for OOD generalization with CLIP models. The method uses a feature prediction network to hallucinate image features corresponding to unknown texts at training time. The resulting training procedure is more robust, since it takes into account synthesized features from unseen classes. The authors also propose a self-distillation mechanism to complement the method.

**Strengths:**

- I found the method interesting and novel.
- I found the paper easy to follow. Section 3.2 and Figure 2 are especially informative and organized very intuitively.
- The method seems like it could be useful.

**Weaknesses:**

- the self-distillation mechanism is easy-to-think-of, but this is okay, since it is not the main innovation.
- My main concern with this paper would be the results. In particular, they are not state-of-the-art (see [Maple] and [Clipood]). Furthermore, the reported CoOp performance seems low. With some tuning, CoOp can be much better, e.g. [KgCoOp] reported a harmonic mean of 74.6 for CoOp on average, compared to 71.7 reported by the authors.  From personal experiments, I know that simply finetuning both encoders along with the prompt with cross-entropy can achieve much better results on these benchmarks, (80.3 % HM on the base-to-novel benchmark, average of 11 datasets). However, the authors seem to focus on just prompt tuning, so it might be ok.

I'm incline to think that that the method is interesting enough for acceptance, even though, in my opinion, the results are not state-of-the-art.

[Maple] Muhammad Uzair Khattak et al. "MaPLe: Multi-modal Prompt Learning"

[Clipood] Yang Shu, Xingzhuo Guo et al. "CLIPood: Generalizing CLIP to Out-of-Distributions"

[KgCoOp] Hantao Yao et al. "Visual-Language Prompt Tuning with Knowledge-guided Context Optimization"

**Questions:**

None.

---

> ### Author Response · Authors · 2023-11-20
> **Response to Reviewer Rp4n**
>
> Thanks for the positive feedback on our work and recognition of its novelty. To address concerns about SOTA results, we include more comparisons in our “Response to common concern: comparing with the SOTA”, where our method is found to consistently improve more recent methods, including the suggested Maple and KgCoOp.

---

> ### Comment · Reviewer_Rp4n · 2023-11-20
> **response to response**
>
> The new results are quite impressive. I raise my score to 8 and advocate for publication.
>
> In my original review I had commented that CoOp is much better than what the authors report if the authors make a good faith attempt at tuning that baseline instead of copying the number from previous publications. This point remains unaddressed, but I believe it should not factor into my review since I know so many papers in this area do not tune baselines.

---

> > ### Author Response · Authors · 2023-11-21
> > **Response to Reviewer Feedback**
> >
> > Many thanks for the positive feedback and score raising!
> >
> > We are right in the middle of experimenting with different finetuning methods other than prompt learning. Specifically, we are looking at the suggested backbone tuning (i.e., image/text encoders) as well as adaptor tuning. These efforts are not only to continue pushing the boundaries of performance (will report more results in Appendix once available), but also to have a better understanding of potentially different overfitting behaviors of various finetuning methods, and more importantly, how OGEN will handle different cases. Furthermore, we plan to release codes to let people in this field contribute to such investigations and pay close attention to the overfitting issue with vision-language model finetuning.

---

### Official Review · Reviewer_Y9rR · 2023-10-29

**Soundness:** 3 good
**Presentation:** 3 good
**Contribution:** 2 fair
**Rating:** 6
**Confidence:** 3

**Summary:**

This paper addresses the limited generalization capabilities of existing vision-language models, which struggle to handle open-domain visual concepts. The authors propose a novel approach called OGEN to improve the out-of-distribution (OOD) generalization of finetuned models. OGEN introduces a class-conditional feature generator that synthesizes OOD features using only the class name of any unknown class, helping to regularize the decision boundary between in-distribution (ID) and OOD data. Additionally, an adaptive self-distillation mechanism is employed to prevent overfitting. Experimental results demonstrate that OGEN achieves considerable improvements in OOD generalization performance across different settings.

**Strengths:**

* The forgetting problem is important in foundation models during fine-tuning.
* The over-fitting observation can support the paper's main claim.
* The proposed method is reasonable.

**Weaknesses:**

I believe it would be beneficial for this paper to include a comparison with relay-based methods, such as sampling a subset from Lioan-5B and using it for replay. The "class-conditional feature generator" seems to serve as a proxy for the replay data, so it would be valuable to directly explore the use of replay methods. As a result, I find the novelty of the proposed approach to be somewhat limited considering this concern.

**Questions:**

See above.

---

> ### Author Response · Authors · 2023-11-20
> **Response to Reviewer Y9rR**
>
> Thanks for the feedback! Below is our answer to your main (and great!) question to hopefully address the novelty concern.
>
> **Compare with relay-based methods, e.g. sampling a subset from LAION-5B and using it for replay.**
>
> We have experimented with replay methods on the older but sufficiently large LAION-400M dataset. The goal is to compare our synthesized OOD features against the real replay data (sampled from LAION-400M) in terms of their impact on training regularization. Please refer to **Appendix-Section D** in the revised paper for experimental details, especially how we sample the replay data, including class filtering, sampling strategy and data size. Our high-level observations from **Fig. 7** are that replay methods perform well with large data size, but suffer from low data efficiency as well as large memory cost (see detailed reasoning in Section D). By contrast, our feature synthesis approach avoids these issues by synthesizing hard OOD features on the fly. One future direction is to combine feature synthesis with replay methods to take advantage of their respective benefits of data efficiency and diversity. We will consider moving some of these discussions/results to the main paper to highlight the novelty and benefits of our feature synthesis approach.

---

> > ### Comment · Reviewer_Y9rR · 2023-11-21
> >
> > Thanks for the effort on new experiments. I raised my score to 6 to acknowledge this.

---

> > > ### Author Response · Authors · 2023-11-21
> > > **Response to Reviewer Feedback**
> > >
> > > Many thanks for raising the score! We will continue to improve the paper by integrating new discussions and results and work towards code release. If you have more questions, please feel free to let us know.

---

### Official Review · Reviewer_sauS · 2023-10-31

**Soundness:** 3 good
**Presentation:** 3 good
**Contribution:** 3 good
**Rating:** 5
**Confidence:** 2

**Summary:**

The paper improves the out-of-distribution (OOD) generalization of vision-language models, especially CLIP, when they are finetuned on downstream tasks. The paper makes the following contributions:

- It reveals the overfitting problem of existing finetuning methods, such as prompt learning, that degrade the OOD performance of CLIP models.

- It proposes a novel method called OGEN, which consists of two components: a class-conditional feature generator and an adaptive self-distillation mechanism.

- The paper evaluates OGEN on various downstream tasks and datasets and shows that it consistently improves the OOD generalization of different finetuning methods for CLIP models.

**Strengths:**

- The proposed method addresses a novel and important problem of improving the OOD generalization of vision-language models, especially CLIP when they are finetuned on downstream tasks.

- The paper proposes a novel method called OGEN, which consists of two components: a class-conditional feature generator and an adaptive self-distillation mechanism. The feature generator synthesizes OOD image features given the name of an unknown class, by extrapolating from the most similar known classes. The self-distillation mechanism uses an adaptive teacher model that is an exponential moving average of past model checkpoints within a local time window. The teacher model guides the student model to avoid overfitting and maintain a good trade-off between in-distribution and OOD performance.

- The paper evaluates OGEN on various downstream tasks and datasets and shows that it consistently improves the OOD generalization of different finetuning methods for CLIP models. It also provides comprehensive ablation studies and analysis to validate the effectiveness of each component of OGEN.

**Weaknesses:**

(1) The proposed method mainly compared with CoOp (IJCV'22), Co-CoOp (CVPR'22), and VPT (ECCV'22).
However, before the deadline of ICLR, the state-of-the-art methods are released here: https://github.com/muzairkhattak/PromptSRC
MaPle (CVPR'23) and  PromptSRC (ICCV'23) need to be discussed in this paper.

(2) In Tab 3 and Tab 4, the proposed class-conditional feature generator slightly decreases the performance of the base classes.
In the appendix Fig 5, there are some explanations regarding the performance increase in the new classes and performance variation in the base classes. These discussions need to move to the main script.

(3)	The first step of OGEN, novel class extrapolation, is problematic. It is unreasonable to utilize base features to extrapolate novel features, since there are usually large conceptual gaps between base and novel classes. The authors provide a special case, that is “cat, bear->raccoon”. But in CIFAR-10, for example, I think the “ship” class is not conceptual close to any other classes.

(4)	Another contribution of this paper, claimed by authors, is Adaptive Local Mean Teacher (ALMT), which I think is just a trivial trick of hyperparameter tuning. The difference between ALMT and conventional MT is just modifying the sliding window size. The novelty is quite low, and the performance improvement brought by ALMT over “No distillation” is insignificant (less than 1%), as shown in Table 6.

(5)	The performance of OGEN is too low and outdated. In existing prompt learning papers in CVPR’23 (such as [1,2]) and ICCV’23 (such as [3,4]), the “New” accuracy in base-to-novel generalization setting is already about 75%, but OGEN can only achieve around 70%.

(6)	The paper writing is quite poor and hard to follow. The objective function is missing in Sec 3.3.

[1] CVPR 2023. MaPLe: Multi-modal Prompt Learning
[2] CVPR 2023. LASP: Text-to-Text Optimization for Language-Aware Soft Prompting of Vision & Language Models
[3] ICCV 2023. Self-regulating Prompts: Foundational Model Adaptation without Forgetting
[4] ICCV 2023. Read-only Prompt Optimization for Vision-Language Few-shot Learning

**Questions:**

My main concern is the baselines selected to compare in this paper are too old (methods published in 2022). MaPle (CVPR'23) and  PromptSRC (ICCV'23) need to be discussed in this paper.

---

> ### Author Response · Authors · 2023-11-20
> **Response to Reviewer sauS**
>
> Thank you for the constructive feedback and helpful suggestions for our manuscript. Regarding the comparison with state-of-the-art methods, please refer to our “Response to common concern: comparing with the SOTA”.  For your other comments, below is our point-by-point response.
>
> **Move Appendix Fig. 5 to the main paper, poor paper writing, and missing objective function in Sec 3.3.**
>
> We appreciate the reviewer for highlighting these presentation issues. We will fix them as suggested, and try to identify and rewrite sections that are dense or difficult to follow as well. If you have other suggestions on areas that could be improved to help us understand where to focus, it would be really helpful.
>
> **Class extrapolation is unreasonable when there are large conceptual gaps between base and novel classes.**
>
> We agree the quality of class extrapolation highly depends on the semantic similarity between base and novel classes. We do observe faithfully extrapolated class examples when the semantic gap is small (Fig. 3 for example), but less faithful ones otherwise.
>
> Among the many downstream datasets we have experimented on, most of them demonstrate shared regularities between classes (unlike CIFAR10), which gives rise to good extrapolation quality. Examples include datasets with a large number of conceptually close classes like FGVCAircraft and StanfordCars, as well as datasets with only a few but pattern-sharing classes like EuroSAT (satellite images) and DTD (texture images). The well-extrapolated OOD data on these datasets yield consistent, and sometimes large, gains on novel class performance, see Fig. 6(a).
>
> On the other hand, there are datasets UCF101 and Caltech101 that have relatively large semantic gap between classes. Their data size is also small (13k and 9k respectively). Both of the two factors will negatively impact the class extrapolation quality. However, we still observe from Fig. 6(a) convincing gains on novel classes for UCF101 and Caltech101. The reason is twofold: 1) the abundance of extrapolated class examples alleviates overfitting on small datasets by scaling up the negative data, 2) extrapolated data from known classes, despite being noisy, can still serve as ``semi-hard'' negatives to regularize decision boundaries. One natural direction for improvement is to enrich the data used for extrapolation, e.g. by combining a living memory of diverse web data with the few training classes on downstream datasets.
>
> **Limited novelty and performance gains of ALMT.**
>
> In response, ALMT is not the main innovation of this paper, and we view ALMT as a surprisingly simple add-on to reduce overfitting. Note behind the adaptive window size in ALMT lies a well-grounded rationale, i.e. to reduce both overfitting and underfitting during model distillation (see texts before Eq. 7). Such MT adaptation does not require extensive hyperparameter tuning to gain improvements, which is non-trivial.
>
> Performance-wise, Tables 3&6 verify that, ALMT in its simple form, can indeed improve both base (ID) and new (OOD) class performance. Notably, the improvement on base class is larger ($ \approx$ 1\%), which is particularly useful in our case since our OOD feature generator may slightly degrade the base class performance when improving OOD generalization. Overall, ALMT serves as an optional booster that helps to strike a good balance between the base and new class performance. We will add such discussions in the revised paper, with ALMT's role to be toned down accordingly.

---

> > ### Author Response · Authors · 2023-11-22
> > **Gentle reminder of the discussion deadline**
> >
> > Dear Reviewer sauS, thanks again for your insightful feedback! Taking into account your tight schedule, we are sending you this gentle reminder that the rebuttal deadline is approaching. Would you mind letting us know if your concerns have been fully addressed or if you have more questions? In the latter case, we will try our best to address any lingering questions before the deadline. Looking forward to your reply! In the meantime, we will be continuously improving the paper following your suggestions and working towards code release. Thanks for your time!

---

### Author Response · Authors · 2023-11-20
**Response to Common Concern: Comparing with the SOTA**

Thanks to all reviewers for the thoughtful comments and suggestions. We have finished experimenting with the following 4 state-of-the-art methods as suggested by different reviewers. We compare performance with and without adding our OGEN approach.

- [SHIP] Wang et al. ``Improving Zero-Shot Generalization for CLIP with Synthesized Prompts'', ICCV 2023.
- [KgCoOp] Yao et al. ``Visual-Language Prompt Tuning with Knowledge-guided Context Optimization'', CVPR 2023.
- [MaPLe] Khattak et al. ``MaPLe: Multi-modal Prompt Learning'', CVPR 2023.
- [PromptSRC] Khattak et al. ``Self-regulating Prompts: Foundational Model Adaptation without Forgetting'', ICCV 2023

We’ve revised the paper and uploaded the new version that includes such new comparisons:
- Detailed results in Table 1, where OGEN uniformly improves the generalization performance on new classes for the 4 SOTA methods
- New discussions in Section 4.1 (red highlighted) about how and why OGEN improves performance, depending on the length of training schedule
- Fig. 5 in Appendix B to exemplify how SOTA methods still suffer from overfitting and how OGEN improves their learning curves.

Please refer to the revised paper for details and let us know if you have any further questions - we are happy to follow up! Thanks for your time!

---

### Meta-Review · Area_Chair_4wLS · 2023-12-02

**Metareview:**

The paper propose a class-conditional feature generator to synthesize OOD features using just the class name of any unknown class (referred to as OGEN). Such synthesized features will provide useful knowledge about unknowns and help regularize the decision boundary between ID and OOD data when optimized jointly. A adaptive self-distillation mechanism is also introduced to regularize the feature generation model during joint optimization. The proposed method is validated on 11 diverse datasets.
Pros:
* OGEN can be applied on various prompt learning based methods
* Average performance over 11 datasets show consistent improvement when applying OGEN on various prompt learning based methods
Cons:
* Novel class extrapolation, is problematic. It is unreasonable to utilize base features to extrapolate novel features, since there are usually large conceptual gaps.
* Some setting performance is not improved. For instance, PromptSRC on DTD dataset.

The authors add experiments on new state-of-the-art like MaPLe and PromptSRC and mostly achieve improvement as expected. Hence, Rp4n raise rating to 8. Reviewers sauS and v4h7 share similar concern but did not response to the rebuttal.
AC thinks OGEN is very useful as it can be applied on  various prompt learning based methods and, in most setting, OGEN shows improvement. Hence, AC recommends for acceptance.

**Justification For Why Not Higher Score:**

Some setting performance is not improved. For instance, PromptSRC on DTD dataset.

**Justification For Why Not Lower Score:**

The authors add experiments on new state-of-the-art like MaPLe and PromptSRC and mostly achieve improvement as expected.

---

### Decision · Program_Chairs · 2024-01-16

Accept (poster)